# Improving Biocontrol Potential of Antagonistic Yeasts Against Fungal Pathogen in Postharvest Fruits and Vegetables Through Application of Organic Enhancing Agents

**DOI:** 10.3390/foods14173075

**Published:** 2025-08-31

**Authors:** Gerefa Sefu Edo, Esa Abiso Godana, Guillaume Legrand Ngolong Ngea, Kaili Wang, Qiya Yang, Hongyin Zhang

**Affiliations:** 1School of Food and Biological Engineering, Jiangsu University, Zhenjiang 212013, China; sefgare@gmail.com (G.S.E.); esaabiso@gmail.com (E.A.G.); 1000005483@ujs.edu.cn (K.W.); 2Department of Agricultural, Environmental and Food Sciences, Università degli Studi del Molise, 86100 Campobasso, Italy; ngolngea@yahoo.fr

**Keywords:** biological control, antagonistic microbe, enhancer, molecular mechanism, fungal pathogen, fruit and vegetable

## Abstract

Fruits and vegetables are essential for a healthy diet, providing vital nutrients and contributing to global food security. Fungal pathogens that interact with fruits and vegetables reduce their quality and shelf life and lead to economic losses and risks to human health through the production of mycotoxins. Chemical fungicides, used to control postharvest pathogens, are posing serious environmental and health risks, driving interest in safer alternative strategies. Biocontrol methods using antagonistic microbes, such as yeasts, are eco-friendly, sustainable, and the most promising, but they often have limited efficacy and specificity in diverse produce. There is growing interest in the innovative enhancement of biocontrol strategies. The present review shows that inducing, enhancing, co-application, encapsulation, and post-application treatments are common enhancement techniques, while environmental, host, and pathogen characteristics, antagonistic microbial traits, and chemical inputs are the major gearing factors for the best application methods. These methods do not involve genetic modification, which is adequate to reduce the proliferation of GMOs (Genetically Modified Organisms) while optimizing antagonistic microbial performance by promoting growth, inducing host resistance, enhancing antifungal properties, improving adhesion, and boosting stress tolerance. Most enhancers fall under groups of nutritional additives, protective carriers, growth stimulants, and encapsulants. Integrating these enhancers and best methods promises reduced postharvest losses, supports sustainable agriculture, and addresses economic losses and food security challenges. This study highlights the role of organic and natural elicitors, their application methods, their mechanisms in improving BCAs (Biological Control Agents), and their overall efficiency. This review concisely compiles recent strategies, calling for further research to revolutionize fungal pathogen management, reduce food waste, and promote responsible farming practices.

## 1. Introduction

Fruits and vegetables are an essential part of a healthy diet. They provide vital nutrients for our health and contribute to global food security. However, this class of foods is affected by huge postharvest losses ranging from 28 to 55% of total production, which translates to an annual loss of approximately USD 750 billion [1,2]. It is estimated that up to 30–50% of harvested produce is lost globally due to postharvest diseases caused by fungal infection [3,4]. Significant research and innovation efforts have focused on reducing F&V postharvest losses [5,6]. The study explores the ways in which harvested fruits and vegetables are prone to decay and quality deterioration during storage. Although traditional packaging and chemical treatments are effective, they are harmful to the environment and human health. Hence, greater requirements for food preservation technology are increasingly proposed.

The primary factors contributing to these losses include suboptimal handling, physical damage, postharvest diseases (fungal and bacterial), insect pests, small animal attacks (rodents, birds, and other animals), harvesting at inappropriate stages of ripeness, insufficient storage and packaging methods, and poor transport conditions [7]. Furthermore, physiological processes, such as respiration, transpiration, and ethylene production, exacerbate the postharvest deterioration of fruits and vegetables [8]. Common diseases in fruits and vegetables arise from fungi, bacteria, viruses, and oomycetes (fungus-like organisms) [3,9], while fungal pathogens take the lion’s share. The most common of these fungal pathogens are *Botrytis cinerea*, *Penicillium* spp., *Aspergillus* spp., *Colletotrichum* spp., and *Alternaria* spp. [3,10]. These pathogens drastically drop the quality and shelf life of postharvest produce, affecting marketability and leading to economic setbacks for producers and processors. They also pose health risks to consumers [3,11].

Management of these fungal diseases takes different forms, with varying mechanisms. Mechanisms broadly encompass physical methods, chemical methods, and biological methods [12,13]. Some of the physical approaches have the benefits of maintaining nutritional quality and offering easy applicability, with an absence of residual effects. The potential for damage to fruits or vegetables, impacts on sensory and physicochemical quality, high energy costs, short-term effectiveness, and regulatory challenges are among the limitations this method [14]. Chemical fungicides are effective and widely used (>90%) in the postharvest disease management of fruits and vegetables. Commercial accessibility, ease of application, and effectiveness in broad product-based applications make this method preferable over others [15]. Despite its benefits, this method lacks the full approbation of the population due to its negative impacts on the environment and human health and its role in the development of resistance among pathogens. The biological strategy has gained attention mainly from a sustainability perspective [16,17]. Biological methods’ strengths are that they have little or no adverse effects on human health, they do not generate resistance in pathogens, they have long-lasting effects, and they affect the physicochemical quality of products as little as possible [18,19]. Other advantages include their environmental friendliness and compatibility with integrated pest management programs [11,20]. Antagonistic yeasts have a promising ability to kill and compete with pathogens without requiring the use of antibiotics that can promote the development of antibiotic-resistant strains in the environment. *Saccharomyces cerevisiae*, *Wickerhamomyces anomalus*, *Debaryomyces hansenii*, *Meyerozyma* spp., *Candida* spp., *Pichia* spp., and *Metschnikowia* are among the common antagonistic yeasts used in biocontrol methods for fruit and vegetable fungal pathogens. They mainly inhibit fungal pathogen growth through various mechanisms, including physical and nutrient competition, parasitism, rapid growth, efficient colonization of surface injuries, and physiological disruption through volatile organic compound (VOC) production [21].

Nevertheless, their effectiveness fluctuates most often under different environmental conditions, which limits their specification and market value, restricting their use worldwide. The overall impact of biological control on managing plant diseases in agricultural settings is currently minimal, accounting for approximately 1% of agricultural chemical usage, due to these limitations [22]. Similarly, one study shows that although the potential of antagonistic yeasts spans crop protection, food safety, and disease prevention, their commercial availability remains limited [23]. An important implication of these findings is that future research should prioritize the optimization and commercialization of yeast-based biocontrol products, effectively bridging the gap between scientific discovery and practical application in combating harmful pathogens.

There are different approaches to making BCAs more efficient, such as the use of mixed cultures, their physiological manipulation, and their combination with different types of substances [24]. Enhancement techniques involve the application of supplements, such as chitosan, phytic acid, ascorbic acid, trehalose, alginate oligosaccharide, beta-glucans, salicylic acid, essential oils, and plant extracts. These approaches improve the efficiency, cell viability and performance, adherence, and fungicidal properties of antagonistic yeasts. Each substance used can variably strengthen the cell walls of antagonistic yeasts, improve their colonization of fruit surfaces, and trigger the synthesis of antifungal compounds [25]. By fine-tuning these enhancement approaches, the full potential of antagonistic yeasts as dependable and sustainable solutions for managing postharvest diseases can be achieved. Many key factors shape the optimization of antagonistic yeasts’ mechanisms. These include environmental factors, such as temperature, relative humidity, UV intensity, pH levels, and the ecology and mechanisms of antagonism [26]; host plant characteristics, including inoculum density, virulence, and host plant genotype [27]; pathogen characteristics, such as its concentration, resistance to antagonistic microbes, and self-defense mechanisms [28]; biocontrol agent properties, like the mode of action, viability, stability, and application timing [18]; and chemical inputs, such as pesticides and fertilizers, which may harm antagonistic microbes or reduce their efficacy [29].

The present review focuses on three key areas; (i) highlighting the importance and necessity of improving antagonistic yeast by identifying and compiling reagents utilized in several studies to improve/boost their performance; (ii) highlighting possible application methods and exploring effective delivery techniques for enhancers along with antagonistic yeasts to mitigate target fungal pathogens; and (iii) compiling different mechanisms of actions by exposing the underlying mechanisms behind the various improvements in antagonistic yeasts against the development of pathogenic fungi. Over the past three decades, hundreds of studies have studied biological postharvest disease management in fruits and vegetables, particularly using antagonist yeasts. These studies confirm that fungal biocontrol is both effective and sustainable. However, existing findings also reveal limitations, including product specificity and low market accessibility due to challenges in mass production. Commercial development of a biocontrol product is a complex process involving several stages, including initial discovery and screening, pilot testing, upscale production and formulation, and patenting and registration, all of which require financial support from public grant agencies or a business partner [21]. Further constraints include the short shelf life and variable efficacy of biocontrol agents (BCAs), which depend on environmental conditions, host cell types, the substances applied, and other chemicals used on host cells.

A literature review was carried out by systematically searching Google Scholar, ScienceDirect, and Google to find peer-reviewed articles on antagonistic yeasts for biocontrol of fungal pathogens of fruits and vegetables. Search terms included “antagonistic yeast,” “biocontrol,” “postharvest disease,” and “fungal pathogen” in combination with specific hosts (e.g., apple, citrus, and tomato). Results were sorted by relevance, publication date (kept to very recent publications as much as possible), and availability of full-text articles. Curation was limited to research elucidating biocontrol mechanisms and efficacy, with the explicit exclusion of any AI-generated content to confirm that all synthesized information came from the published scientific literature.

In conclusion, there is no doubt that most antagonistic yeasts are effective in the biocontrol of specific pathogens in specific environments and under specific conditions, as demonstrated by numerous studies worldwide. However, without improving the viability, performance, and antagonistic efficiency of these microbial biocontrol agents, it will be impossible to meet market demand or compete with traditional chemical methods, which have already dominated the global market despite their numerous side effects.

## 2. Causes of Postharvest Loss in Fruits and Vegetables

The characteristics of fruits and vegetables, including their high water content and nutritional content, soft skin, and broad surface area, make them susceptible to various postharvest issues. Akanbi [7] categorizes postharvest losses in these forms of produce as stemming from physiological and technological origins, encompassing deterioration caused by biological/microbiological agents and mechanical damage. Additionally, postharvest losses in fruits and vegetables can come from socioeconomic factors, including a lack of policies promoting efficient use of human, economic, technical, and scientific resources to reduce losses [30,31]. This underscores the fact that despite its significance, the substantial postharvest waste of these commodities has not received sufficient emphasis in research and practice. Alegbeleye [32] reported that fruit and vegetable spoilage represents a significant food issue that has received substantially less research attention compared to produce-related foodborne illnesses and processing technologies, despite its considerable socioeconomic, food quality, and food security implications. Other contributing factors are insufficient resources for developing loss prevention programs, limited knowledge of preservation and distribution technologies, ineffective commercialization systems, inadequate government involvement in commodity postharvest loss management and marketing, and lack of appropriate credit policies tailored to national and stakeholder needs. About 95% of resources are invested into crop production, whereas as low as 5% of this investment is channeled into postharvest preservation [33,34].

Alegbeleye [32], in their study, found that infection and microbiological spoilage of fruits and vegetables may be initiated by various well-characterized microorganisms, including bacteria, fungi, yeasts, and molds. The authors further noted that spoilage microorganisms may be introduced through the farm environment, postharvest handling, processing operations, and various agronomic and environmental factors that predispose produce to microbial deterioration.

Postharvest fruits and vegetables can also face significant challenges due to their inherent properties, making them highly susceptible to microbial infections, pest infestations, physiological disorders, and physical or mechanical damage. These issues are exacerbated by a lack of coordinated effort among key stakeholders, such as producers, traders, processors, retailers, consumers, researchers, and policymakers, to address postharvest loss management effectively. Without collaborative action and shared consideration, the preservation and quality of these perishable commodities remain at risk, highlighting the need for a unified approach to overcome these challenges [3,35].

## 3. Common Postharvest Diseases in Fruits and Vegetables and Fungal Disease Management

Fruits and vegetables are susceptible to different postharvest diseases that can result in considerable losses during postharvest handling, such as harvesting, packaging, transportation, storage, and processing. Most fruits and vegetables have very high moisture content (approximately 70–95% water). They are usually large (0.5–5 kg) and exhibit a higher respiration rate, and they usually have a soft texture, all of which favor the growth and development of several diseases caused by microorganisms between the periods of harvest and consumption [8,9,36].

The most common diseases in postharvest fruits and vegetables are fungal diseases, such as gray mold (*Botrytis cinerea*), blue mold (*Penicillium* spp.), anthracnose (*Colletotrichum* spp.), alternaria rot (*Alternaria* spp.), sour rot (*Geotrichum candidum)*, and *Fusarium* rot (*Fusarium* spp.); bacterial diseases, such as soft rot (*Pectobacterium carotovorum*, *Dickeya* spp.), bacterial spot/blight (*Xanthomonas* spp.), crown gall (*Agrobacterium tumefaciens*), and pseudomonas rot (*Pseudomonas* spp.); viral diseases, such as tomato spotted wilt virus (TSWV), cucumber mosaic virus (CMV), and potato virus y (PVY); and oomycete diseases (*fungus-like organisms*), such as late blight (*Phytophthora infestans*) and phytophthora rot (*Phytophthora* spp.) [3,9,16]. These pathogens can cause various types of rot and spoilage, making them unsuitable for consumption and potentially producing harmful mycotoxins [37,38].

It is widely recognized that fungi are primarily responsible for postharvest losses in these produce items [3,9,39]. Different mechanisms are used by pathogens in the infection of host cells. According to the literature [40,41,42], fungal pathogens in fruits and vegetables can be classified into different groups based on their characteristics during infection. These are neutrophilic fungi, toxigenic fungi, facultative saprophytes, opportunistic fungi, and latent fungi. Figure 1 shows common types of fungal pathogens along with major pathogens and targeted fruits and vegetables.

Proteomic studies reveal pathogen-induced changes in fruit proteomes, including the upregulation of defense-related proteins and the suppression of metabolic pathways crucial for pathogen survival [43,44,45]. After the survival of the pathogen, it then goes on to invade the host cell to cause disease, while some of them are responsible for the production of secondary by-products, such as patulin, citrinin, and penicillic acid, which are predominantly caused by *Penicillium* species, while patulin can also be produced by some *Aspergillus* species. These mycotoxins pose significant health concerns and economic threats, particularly in apples and apple-based products [46]. Fungal pathogens are responsible for the spoilage of various fruits and vegetables [47,48,49]. Table 1 indicates the common fungal pathogens identified in this study.

The quality of fresh produce is influenced by biological processes that are also responsible for its degradation. These biological factors include respiratory rate, germination, root formation, alterations in color, taste, and texture, nutritional changes, ethylene generation, moisture loss, and pathological decay [51,52]. To address these diverse forms of spoilage, numerous researchers have explored multiple strategies for prevention, control, and elimination over prolonged durations [53]. Postharvest fungal pathogens often invade fruits and vegetables after harvesting, which leads to huge losses before reaching consumers [54,55]. Studies show that even after harvest, fruits and vegetables remain active in metabolic processes, which makes them susceptible to decay during postharvest operations [3,56]. One study delved into the various biological and environmental factors that reduce product quality, especially focusing on fungal infections. Nassarawa [57] classified the predominant and well-established approaches of fungal disease management for fruits and vegetables into three primary categories: physical, chemical, and biological methods.

### 3.1. Physical Methods

Various studies have shown that different physical methods for postharvest fruit and vegetable disease control have benefits over synthetic chemical fungicides in maintaining nutritional quality and sensory quality parameters, with an absence of residual effects and easy applicability [58,59].

De Chiara [60] reported that many traditional postharvest technologies can negatively impact the quality of fresh produce and/or affect the sustainability of processes, while a recent study spotlighted emerging innovative physical techniques like vacuum and hydrocooling, microwave heating, pulsed electric field, high hydrostatic pressure, and cold plasma application designed to reduce nutrient losses and enhance consumer acceptability. In that study, they also stated that each technique offers unique advantages and limitations. In contrast, there are some limitations to this approach due to varying efficacy, potential for damage, high costs, short-term effectiveness, and regulatory challenges [61].

### 3.2. Chemical Methods

Chemical methods for postharvest loss management of fruits and vegetables include other techniques that are used broadly in the world. This method has several advantages and limitations. According to Wu [62], synthetic fungicides have been proven to significantly reduce the incidence of postharvest diseases caused by various fungal pathogens. Studies have also demonstrated that synthetic fungicides can have negative effects on biodiversity [63]. According to that study, out of the 13 studied components of agricultural intensification, use of pesticides, especially insecticides and fungicides, had the most consistent negative effects, such as decline in species diversity of plants, impacts on carabids and ground-nesting farmland birds, and effects on the potential for biological pest control. Specifically, they stated that bird species’ diversity declined with increasing frequency of fungicide application. The misuse of artificial fungicides undermines the safety of farm crops because of their lasting effect on human health and the environment and susceptibility to pathogen resistance [64].

### 3.3. Biological Control Methods

The third and most recent method, which has received attention from producers, processors, consumers, and researchers over the past three decades, comprises biological disease control methods for postharvest fruit and vegetable decay. Biocontrol has numerous advantages over the other two methods in that it is more natural and has no residual effects and pathogens do not develop resistance to it. This approach utilizes various natural substances, including plant-derived extracts and essential oils, along with competitive bacterial and yeast strains, whereas actinomycetes have emerged as a novel focus in recent studies [65]

Different studies have tried to identify working principles of biocontrol methods against fungal pathogen infections in postharvest produce. As Ling [66] reported, endophytic fungi—living harmlessly inside of plant tissues—generate antifungal volatile organic compounds (VOCs). These airborne substances show significant effects in postharvest disease control due to their non-toxic nature and dispersive properties, which make them viable biological control alternatives.

Other studies have revealed that environmental yeasts exhibit significant biocontrol potential against pathogens [17,25]. These microorganisms apply multiple antagonistic strategies, including nutrient reduction, parasitic activity, production of antifungal metabolites, and stimulation of plant defense systems. This includes the role of reactive oxygen species (ROS) in mediating protective responses. Meaning they are highly reactive, oxygen-containing molecules that can damage important cellular components like DNA, proteins, and lipids

Similar findings by Li [67] showed that *Bacillus* spp., especially *B. subtilis*, serve as efficient biocontrol organisms. Their effectiveness is based on competitive hindering, synthesis of antimicrobial compounds, and activation of plants’ systemic resistance. Application can occur during either pre-harvest or postharvest stages, with endophytic alternatives providing greater protective benefits.

#### Common Biocontrol Agents for Postharvest Fruits and Vegetables

Recent advances in biotechnological approaches have positioned biological control as a widely researched, environmentally friendly solution, employing beneficial microbes to combat postharvest phytopathogens as an alternative to traditional chemical treatments [68]. These biocontrol agents are applied as postharvest treatments, either alone or in combination with other methods, to extend shelf life and reduce spoilage caused by fungal pathogens. Their use aligns with the growing demand for eco-friendly and safe food preservation strategies [69]. Table 2 shows a list of common biocontrol agents studied at different times to combat fruit and vegetable postharvest fungal pathogens.

## 4. Factors Affecting the Efficacy of Antagonistic Microbes

### 4.1. Environmental Factors

Environmental factors, such as temperature and humidity, have significant effects on the growth and activity of antagonistic microbes [100]. The antagonistic efficacy of almost all yeast, bacteria, and fungi is highly dependent on the optimal temperature and humidity levels of the environment. pH can strongly affect the survival and functionality of antagonistic microbes. Chowdhury [101] reported that under acidic conditions, some bacteria, such as *Bacillus amyloliquefaciens* MBNC, can retain their antagonistic efficiency against fungal phytopathogens, although their metabolite production is altered. In addition, light exposure can influence the activity of these microbes. UV light harms microbes and reduces their viability. It was also reported that understanding ecological aspects and the mechanisms of antagonistic fungi is essential for developing effective and sustainable biocontrol strategies against fungal pathogens [26,102]. All of these environmental limitations of BCA can be overcome by enhancing antagonistic microbes. The study by Magan [26] supports this in reporting that extreme climatic events further impact the efficacy of BCAs and may require systems that can identify more resilient candidate BCAs for fungal pathogens and pest control in the near future.

### 4.2. Host Plant Characteristics

Plant species and variety determine the efficacy of antagonistic microbes. Pathosystem (conceptual framework that define and describe the entire system of interactions between host cell and pathogen in our case fungal pathogen within a given environment) components, such as inoculum density and virulence, and host plant genotype, are important factors influencing the efficacy of plant disease biocontrol [27]. Plant health influences these antagonistic microbes’ activity. Stressed or unhealthy plants may not effectively support microbial activity. Some antagonistic microbes trigger the plant’s own defense mechanisms, which can vary in effectiveness depending on the health and genetics of the plant. Therefore, understanding the complex interplay between plants, beneficial microbes, and pathogens is crucial for the development of effective and sustainable disease control strategies in agriculture [28,103,104]. Antagonistic microbes can suppress postharvest diseases in fruits and vegetables through various mechanisms, but their efficacy depends on factors like maturity, storage conditions, and combinations with other treatments [105]. Nonetheless, enhancing antagonistic yeast will inhibit these challenges.

### 4.3. Pathogen Characteristics

Fungal species and their strains can also determine the efficacy of antagonistic microbes, as some fungi are more resistant to antagonistic microbes than others and vary in their pathogen loads. High concentrations of fungal pathogens can overcome antagonistic microbes. Insight into the self-defense mechanisms used by pathogens to hinder antagonistic attack offers a novel strategy for improving the durability of biologically based disease control practices and is applicable to the introduction of transgenes [28]. Different studies have been working on better understanding pathogen–antagonist–host interactions and improving biocontrol activity to establish microbial antagonists as viable alternatives to synthetic fungicides in postharvest disease management [106,107,108]. By enhancing antagonistic microbes, it is possible to make it withstand resistance from the pathogen.

### 4.4. Chemical Inputs

Pre- and postharvest chemicals, such as pesticides and fertilizers, can harm antagonistic microbes or reduce their effectiveness. Supplemental organic inputs like compost and similar materials can stimulate beneficial microbial populations. Synthetic agricultural chemicals, such as pesticides and synthetic fertilizers, frequently inhibit the development, reproductive capacity, and viability of beneficial microorganisms used for biological control, consequently diminishing their efficacy [29].

### 4.5. Biocontrol Agents’ Characteristics

Strain selection of antagonistic microbes is also essential for their efficacy. Selecting the correct strain is critical [109]. The mode of action also affects efficacy, as some microbes produce antifungal compounds, while others compete for resources or induce plant resistance; as such, the mechanism matters [110,111]. The ability of microbes to survive storage and application and on the plant’s surface is crucial, and the application time of microbial antagonists affects its efficacy, while postharvest applications are more effective than pre-harvest applications.

The new paradigm shift proposed in biocontrol research includes the integration of low-risk chemical fungicides, natural antimicrobial substances, and physical means, such as hot water treatment, irradiation with ultraviolet light, microwave, and infrared treatment in the biocontrol process, as well as enhancement of the expression of crucial biocontrol genes and/or combining genes from different biocontrol agents in mass production, formulation, storage, or in response to exposure and contact with host plant tissues after application.

In general, different studies have been conducted to improve the efficiency of antagonistic yeasts by supplementing them with enhancers, such as trehalose, ascorbic acid, chitosan, alginate oligo saccharides, or certain elicitors, during yeast culture or shortly before their use [112,113]. The supplements enhance yeast cells’ efficacy against several stress factors, including temperature fluctuation, dehydration, ultraviolet irradiation, and infection by pathogens [113]. In some studies, it has also been observed that trehalose is especially good at safeguarding cell structure; it consolidates the membrane structure of antagonist strains and improve their properties to protect cell content at the time of application and against protein damage under stress [53,114,115].

## 5. Common Reagents Used as Enhancers of Antagonistic Yeast and Their Characteristics

Common reagents used can be classified into four major groups: nutrient supplements and molecules, complex structure polysaccharides, antioxidants, and pH modulators [116,117].

### 5.1. Nutrient Supplements and Molecules

Nutrients play a crucial role in enhancing the efficacy of biocontrol agents against plant pathogens. Several studies have demonstrated that supplementing biocontrol agents with nutrients improved the efficacy of biocontrol. Nutrients affect the interactions between microbial populations and the development of plant pathogens and may play a role in biocontrol [118]. For example, application of organic nitrogen enhanced biomass and antifungal compound production by *Gliocladium virens* and Trichoderma spp., antagonists of *Sclerotium cepivorum*. Another study by Lima [117] revealed that nutrient supplements, such as sugars and nitrogen sources, are required to supply the necessary energy and structural components for yeast growth and metabolic processes.

### 5.2. Complex Structure Polysaccharides

Defensive substances like chitosan and alginate also facilitate yeast’s attachment to production surfaces and induce protective biofilm formation, thereby prolonging their active duration [25,119]. It was reported that cell-free filtrate from *Cryptococcus laurentii* cultured in chitin-enriched medium exhibits high chitinase activity, which has direct antifungal activity against *P. expansum* in pear fruit wounds [120]. A study by Zhang [121] indicates that natural decay of strawberries induced by *R. stolonifer* and *B. cinerea* has been reported to be significantly reduced by using the yeast *R. mucilaginosa* grown in NYDB added with chitosan. In that study, two-dimensional electrophoresis coupled with identification through MALDI-TOF/TOF analysis indicated that certain proteins associated with fundamental cellular metabolism had higher levels in the yeast, which may be responsible for enhanced biocontrol efficacy. Pathways like MAPK signaling, peroxisome, glutathione metabolism, and oxidative phosphorylation could have been the cause of increased biocontrol activity of *R. mucilaginosa* induced by chitosan [122].

Other defensive compounds, like trehalose and glycerol, significantly enhance yeast survival under both temperature stress and dehydration stress. Meanwhile, fungal-cell wall-degrading enzymes (e.g., chitinase, glucanase) weaken the structure of the pathogen, rendering it more vulnerable to antagonism by the yeast. This combination of stress-resistant and lytic enzymes constitutes a complete strategy for the optimization of biocontrol yeast efficiency in postharvest application [25,112,122,123]. Chantrasri [124] developed a proteomic method to study how peach fruits gain improved resistance through *Pichia membranefaciens* and salicylic acid treatments. They report that yeast and salicylic acid treatments boosted PR-9 and PR-10 protein activities in peach fruit, which play a central role in the defense mechanism against a range of pathogens.

### 5.3. Antioxidants

Amino acids, such as glutamine, serve a two-way function by improving stress resistance and inducing antifungal compound biosynthesis [125]. It was reported that the expression of some proteins associated with stress response and regulation, e.g., serine/threonine protein kinase, was upregulated by elicitors. Moreover, antioxidants and food types like glutathione, ascorbic acid, amino acids, and phytic acids protect the yeast cells from oxidative damage and thus promote survival under the harsh conditions often experienced within postharvest environments [33]. Antioxidant molecules, such as trehalose and glutathione, are beneficial in boosting antioxidant activity in yeast [126,127]. Lu [128] discovered that antioxidant enzyme activity, that is, catalase (CAT) and superoxide dismutase (SOD), of the test group (yeast cultivated in NYDB with the supplementation of elicitor) was significantly greater than that of the control group (yeast cultivated in NYDB with no additives). Thus, they concluded that supplementation with the elicitor increased the antioxidant capacity of yeast.

### 5.4. pH Modulators

pH modulators like phytic acid are able to modify the microenvironment to favor yeast viability and, at the same time, suppress fungal growth [112]. All of these substances have a synergistic effect of improving the invasion and persistence of yeast and the suppression of pathogens. Further enhancement is provided by plant-derived materials, essential oils, and plant extracts, which not only improve yeast’s performance but also contribute direct antimicrobial activity. Meanwhile, encapsulation methods, such as alginate matrices or chitosan nanoparticles, have twin advantages, as they protect yeast cells from stress conditions and provide controlled delivery of active substances [129,130]. In general, such targeted advancements reduce the limitations of the application of yeast biocontrol, enhancing its efficiency for practical applications. Table 3 shows these enhancing substances and their corresponding biocontrol agent (BCA) subgroups. Difficulties using biocontrol approaches are largely related to specificity, as pathogens require specific modes of action instead of antagonistic microbes [25,112]. To overcome this gap, current research has been directed at increasing the effectiveness of microbial biocontrol agents (MBCAs) using various approaches [131].

## 6. Application Methods for Enhancers of Antagonistic Yeast Performance

Performance enhancers are a range of materials and environmental conditions that enhance the biocontrol efficacy of yeasts. There are different application methods for these enhancers, and selection of these application methods depends on the natural characteristics of antagonistic yeast, postharvest fungal pathogens in fruits and vegetables, the enhancing reagents to be used, environmental conditions, desired outcomes, and utilization purposes [116,142]. They consist of nutritional additives, protective carriers, growth stimulants, and encapsulants [143]. Using performance enhancers has recently gained interest. These strategies are extremely crucial to achieve maximum biocontrol efficacy by promoting improved microbial survival, improved metabolic activity, and improved functional capacity. Various studies show that antagonistic yeast enhancing substances can be applied through different techniques. These application techniques are classified based on the natural characteristics of the constituents and their effectiveness in improving antagonistic yeasts. There are five major categories of antagonistic yeast enhancers, eliciting different application approaches [112,113,123,144]. Figure 2 shows the common roles of enhancers and their interactions with pathogens, antagonistic yeasts, the environment, and host cells.

### 6.1. Pretreatment of Yeast Cells

Pretreatment/inducing involves adding performance-boosting substances to antagonistic yeast cultures before their application. This pre-enhancement approach essentially prepares the yeast cells by improving their stress tolerance and biocontrol activity in real-world conditions. While not traditional genetic modification, it modifies yeasts’ capabilities through physiological conditioning rather than DNA alteration. According to a study conducted by He [123], precultured or induced yeasts increase the induced systematic resistance and enhanced production of ROS in host cells, and they also compete for space and nutrients, play a role in the synthesis of antimicrobial compounds, such as enzymes, volatile compounds, and antibiotics, act through mycoparasitism, secrete lytic enzymes, contribute to biofilm formation, and provide quorum sensing against pathogens.

Ming [145] investigated how mannitol and sorbitol inducing/pretreatment affects stress tolerance in *Debaryomyces hansenii*. Their study revealed that antagonistic yeast cells induced/pretreated with polyol (MT or ST) maintained ~70% survival rates compared to untreated (NT) cells when exposed to oxidative stress (30 mM H_2_O_2_) or heat stress (40.5 °C) for 30 min. While NT cells under such conditions were ~50% viable, pretreated cells had significantly higher survival. Their study revealed that populations of yeasts show an improved ability to compete in ecological niches and thus limit the use of synthetic fungicides. These results establish that mannitol or sorbitol induction significantly improves yeast cell viability in stressful conditions. Chen [138] showed that *Scheffersomyeces spartinae* W9 cultures brought up in nutrient yeast dextrose broth (NYDB) supplemented with β-glucan had greater invasion rates in strawberry wound spots than control cultures developed from the usual NYDB medium. The population of the β-glucan-induced *S. spartinae* W9 increased from 6.1 log10 CFU wound^−1^ to 7.7 log10 CFU wound^−1^ at 24 h, which was significantly higher than that of *S. spartinae* W9 cultured in NYDB (*p* < 0.05). Liu’s [146] study shows that sublethal oxidative stress exposure increases *Candida oleophila*’s tolerance to subsequent stresses and improves its biocontrol performance against fruit pathogens. These enhancements are associated with the upregulation of antioxidant genes and reduced accumulation of reactive oxygen species. Similarly, a study by Zhang [116] shows that inducing improved yeast’s ability to compete with pathogens and enhanced its invading ability over host surfaces, making it an effective and biodegradable tool for replacing synthetic fungicides in postharvest loss management.

This inducing process works by creating selective pressure on yeast populations through nutrient and space competition [25]. Under these constrained conditions, less robust yeast cells struggle, while only the fittest survive. These stress-adapted cells subsequently demonstrate enhanced biocontrol performance, showing high resistance during pathogen interaction and more effective pathogen reduction [123,147].

Pretreatment primes yeast cells to resist and compete effectively against pathogens at the application site. The priming method attains two main benefits: it optimizes both improved cell viability and enhanced functional activity. Enhancing yeasts before exposure optimizes metabolism and resistance to stress by pathogens and is used for improving efficiency. In general, enhancer pretreatment presents a strategic approach to optimize the performance of antagonistic yeasts within agricultural biocontrol [112].

### 6.2. Co-Application of Yeast with Enhancers

Co-application mixing enhancer substances with yeast suspensions prior to treatment delivers both components (yeast and enhancers) simultaneously, providing immediate support for yeast viability and functionality [148]. The purpose of combining these substances with antagonistic yeast during application is to enhance the yeast’s viability and functionality, enabling it to endure reactions from both the host cell and the pathogen. Additionally, these elicitors serve as nutrients for the yeast, further supporting its mechanisms, such as nutrient and space competition, mycoparasitism, host resistance induction, and the production of volatile organic compounds (VOCs) and toxins to combat pathogens [116]. This method combines nutrients like phytic acid, ascorbic acid, amino acids, sodium bicarbonate, and calcium, defensive compounds, like chitosan or bioactive stimulants in some plant extracts, and plant hormones, like auxin indole acetic acid (IAA), Gibberellic acid (GA3), N6-benzyladenine (6-BA), harpin, and trehalose, with antagonistic yeasts to improve their biocontrol efficiency [6,33]. It was also reported that chitosan’s inclusion not only promotes yeast’s adhesion to the surfaces of plants but also its own antimicrobial ability [33,121]. Lima [117], in their study, reported that co-delivery of the elicitor proves to be an easy and effective method for the improvement of agricultural efficiency of biocontrol yeast. They reported that the utilization of food-grade additives could dramatically increase the activity of yeast-based formulation(s) and turn a moderate antagonist, such as the biocontrol isolate LS11, into a highly effective agent. The core benefits of such a process involve the ease of usage and the effect of fastening the action. With the provision of the required supporting compounds concurrently with the yeast, the cells instantly gain an edge in colonization, metabolic activities, and inhibitory ability toward pathogens [149,150]. In this method, combinations of yeast enhancers are applied directly to fruit and vegetable crops for postharvest disease management. Co-application simultaneously provides immediate microbial support, and this results in quicker and more effective control of the pathogens. Different studies have revealed that co-application significantly improves the efficiency and survival of yeast in many applications [108,123,151]. A key limitation of this method is the difficulty of clearly distinguishing whether the elicitors enhance the antagonistic yeast’s efficiency or whether they are involved in the action of directly combating pathogens.

### 6.3. Encapsulation or Immobilization

This method uses defensive coatings to improve the viability and performance of yeast. The physical barrier protects the cells from environmental harshness, including high and low temperatures, ultraviolet light, and drying. Along with providing protection, the medium supports the production of enhancers, thereby providing prolonged nutrition during storage and application stages. Bevilacqua [152] reported that alginate beads with sugars or plant extracts maintain yeast’s metabolic activity and improve antimicrobial activities. One of the first parameters reported in that study, encapsulation yield, was between 87.70 and 108% for different strains. Encapsulation approaches like these effectively improve the stability and efficiency of antagonistic yeasts’ biocontrol [153,154]. Encapsulation of the cells in such protective substances causes the survival rates to significantly improve even under harsh conditions [155,156,157].

*Saccharomyces cerevisiae,* which has an average diameter of about 5 μm, is now a popular option as an encapsulation carrier as whole cells, along with their derivatives, are used effectively as encapsulation agents [135]. It was also found to function as a carrier in the encapsulation process in the early 1970s. Some of the other strains that have also been utilized for microencapsulation are *Torulopsis lipofera*, *Saccharomyces bayanus*, *Endomyces vernalis*, and dairy yeasts, such as *Candida utilis* and *Kluyveromyces fragilis*. *S. cerevisiae* can be said to be an ideal, applicable, and familiar container in the encapsulation method that is utilized to microencapsulate a number of bioactive compounds. In a study carried out by [158], the non-plasmolyzed (NPYC) and plasmolyzed (PYC) *S. cerevisiae* yeast cells were utilized as microcarriers to encapsulate thymoquinone (black cumin seed oil). The study revealed that the bioactivity of thymoquinone was better preserved in PYC, whereas the PYC samples exhibited the lowest thymoquinone degradation ratio (52.63%) during storage. Dadkhodazade [159], in their research, analyzed the encapsulation efficiency of *S. cerevisiae* cells for cholecalciferol (vitamin D3). They reported the maximum EE (76.10 ± 6.92%) at the initial concentration of vitamin D3 of 2.5 mg/g of yeast cells for spray-dried and plasmolyzed yeast cells. The eukaryotic structure and plasma membrane of the cell wall make it distinctive as a physical barrier to environmental factors like oxygen or free radicals and radiation from light. This indicates that encapsulation is used to improve the viability of antagonistic yeasts by protecting them from environmental stresses.

In agricultural applications, microencapsulation systems based on biopolymers are increasingly being utilized to provide beneficial microbes with prolonged shelf life and timed-release properties [160]. This new method addresses major limitations of conventional formulations, with remarkable enhancement of microbial viability and activity under field conditions [155]. The key concerns raised by this approach are fear of chemical interactions between encapsulants, enhancers, and antagonistic yeasts, the simplicity and cost-effectiveness of the technique, and, most importantly, non-target effects on beneficial microbes or host organisms, which require rigorous assessment.

### 6.4. Post-Application Treatment

The post-application approach is a process in which enhancers are applied to the target environment after the initial introduction of yeast, which gives continuous support to maintain biocontrol activity [161]. This method utilizes multiple mechanisms, with enhancers prolonging yeast viability and metabolic activity, improving stress tolerance, indirectly suppressing pathogens, enabling delayed release of compounds, and fostering synergistic interactions for localized biocontrol effects. A practical application like foliar spraying of nutrient solutions (e.g., sugar or amino acid supplements) onto yeast-pre-treated crops is common under this method. Foliar spraying is a very common and straightforward method where nutrients or chemicals are dissolved in water and sprayed directly onto a plant’s leaves, and nutrient solutions are special supplements designed for quick energy. Sugar provides an immediate source of carbon and energy, while amino acids are the basic building blocks of proteins. Providing them directly saves the plant the energy needed to make them from scratch. These treatments keep yeast populations stable by stimulating metabolic function, colonization capacity, and pathogen suppression, which is particularly beneficial in environments with limited resources where indigenous nutrients are depleted [162].

Next to this treatment, interventions are unparalleled in their flexibility, allowing for tailored support based on environmental adjustments or different stages of yeast activity. A study by Sui [112] revealed that one benefit is enhanced yeast stress tolerance, a primary driver of commercial feasibility considering adverse abiotic conditions. The primary challenge of this approach lies in determining optimal application timing. Applying enhancers too early may compromise their supportive capacity for the antagonistic yeast, while delayed application becomes ineffective, as the yeast may have already lost its viability against pathogens, environmental stressors, and host defense responses.

## 7. Mechanisms Involved in Enhancement

As described in earlier studies, enhancers primarily function through mechanisms that include the stimulation of yeast growth, host resistance induction, the enhancement of antifungal activities, adhesion, and stress tolerance [25,116]. The working approach above includes groups of common substances used as antagonistic yeast boosting agents, which fall under five key mechanisms. The schematic diagram presented in Figure 3 illustrates some of these typical mechanisms through which enhancers function to enhance the efficacy of antagonistic yeast in fungal pathogen management.

### 7.1. Stimulating Yeast Growth and Fitness Activity

Enhancers can supply nutrients or growth-stimulating substances that can enhance the growth and metabolic activity of antagonistic yeast in postharvest fruits and vegetables [117,163]. For instance, the inclusion of simple sugars (e.g., glucose) or nitrogen sources (e.g., amino acids) can stimulate yeast’s growth, allowing these microorganisms to compete better with fungal pathogens for resources on the surfaces of stored produce. Moreover, chitin and chitosan compounds are able to induce the formation of yeast enzymes (e.g., chitinases) that break down the cell walls of fungal pathogens, a property that is especially useful for inhibiting infection during storage. By stimulating the growth and activity of yeast, these substances keep the surface of fruit and vegetable crops healthy with a protective layer, thereby preventing spoilage and increasing their shelf life [164,165].

### 7.2. Improving the General Induction of Host Resistance

For fruits and vegetables, biocontrol enhancers can act as elicitors to activate or enhance the innate defense mechanisms of produce against fungal pathogens. Compounds like chitosan, salicylic acid, and β-glucans can be applied directly to harvested produce, triggering defensive responses, such as the production of phytoalexins, pathogenesis-related (PR) proteins, and reactive oxygen species (ROS) [166,167]. For example, a chitosan coating on strawberries or apples can initiate the natural defense of the fruit, making it more resistant to fungal infection when stored. This induced resistance increases the efficacy of the antagonistic yeast, giving a double-layered protection system. By enhancing the defense mechanisms of the host, these elicitors bring about the reduction of spoilage, extend the shelf life, and maintain the quality of postharvest vegetables and fruits [168,169].

### 7.3. Enhancing Antagonistic Properties

Enhancers can increase the production of antifungal metabolites by antagonistic yeast, such as killer toxins, antibiotics, or VOCs, in postharvest fruits and vegetables [66,82]. For example, small amounts of natural compounds, such as essential oils, or specific nutrients can act synergistically with yeasts’ action, thus enhancing their ability to inhibit fungal growth in stored agricultural commodities. These metabolites, according to reports, directly antagonize pathogens by disrupting their cell membranes, inhibiting spore germination, or suppressing hyphal growth. By enhancing the antagonistic activity of the yeast, these compounds make the biocontrol agent effective in inhibiting fungal infection in storage, reducing spoilage, and maintaining the quality of produce [82,116,117].

### 7.4. Improving Adhesion and Colonization

Enhancers can increase the production of antifungal metabolites by antagonist yeasts, such as killer toxins, antibiotics, or VOCs. For example, sub-inhibitory concentrations of certain fungicides or natural compounds (e.g., essential oils) can synergize with yeasts’ activity, augmenting their inhibitory action on fungal growth. By disrupting their cell membrane, inhibiting spore germination, or disrupting hyphal growth, these metabolites have a direct action on pathogens. These substances significantly increase the activity of biocontrol agents in mitigating fungal infections by amplifying the antagonistic character of the yeast [170,171].

### 7.5. Improving Stress Tolerance

Fruits and vegetables are exposed to environmental stresses like temperature fluctuations, low humidity, and ultraviolet radiation throughout storage and transportation during and after harvest. Enhancers can make antagonistic yeasts more resistant to such stresses and preserve their viability and functionality. For instance, osmoprotectants like trehalose or glycerol can make yeast cells more resilient to adverse storage conditions, enhancing their viability and functional efficacy [112,145]. Additionally, antioxidants have the ability to protect yeast against oxidative stress caused by UV exposure during transport. These compounds ensure yeast’s efficacy in controlling fungal pathogens along the postharvest chain, maintaining product quality and reducing losses by enhancing stress tolerance.

## 8. Conclusions and Predictions

Biocontrol methods aimed at fungal plant pathogen control have been attracting consumer interest due to their naturalness, harmlessness to human health, and environmental safety. Their extensive application is, nonetheless, hindered by reduced effectiveness, product specificity, variability in different environmental conditions, short shelf life, limited mass production, less accessibility for market demand, and lack of awareness of their action mechanisms. Overcoming these limitations is critical in order to harness the full potential of biological control in reducing global food wastage. Enhancing the efficacy of antagonistic microbes has emerged as a viable option for overcoming such limitations. An array of organic substances and natural constituents, such as nutrient supplements, antioxidants, and growth regulators, has proven successful in improving the efficacy of biocontrol techniques. This approach not only aligns with consumer preferences for environmentally friendly options but also addresses the immediate need to reduce economic losses and ensure food security. Future research should shift its focus from understanding the mechanisms behind yeast’s efficacy to addressing the practical challenges of product formulation and large-scale commercialization.

## Figures and Tables

**Figure 1 foods-14-03075-f001:**
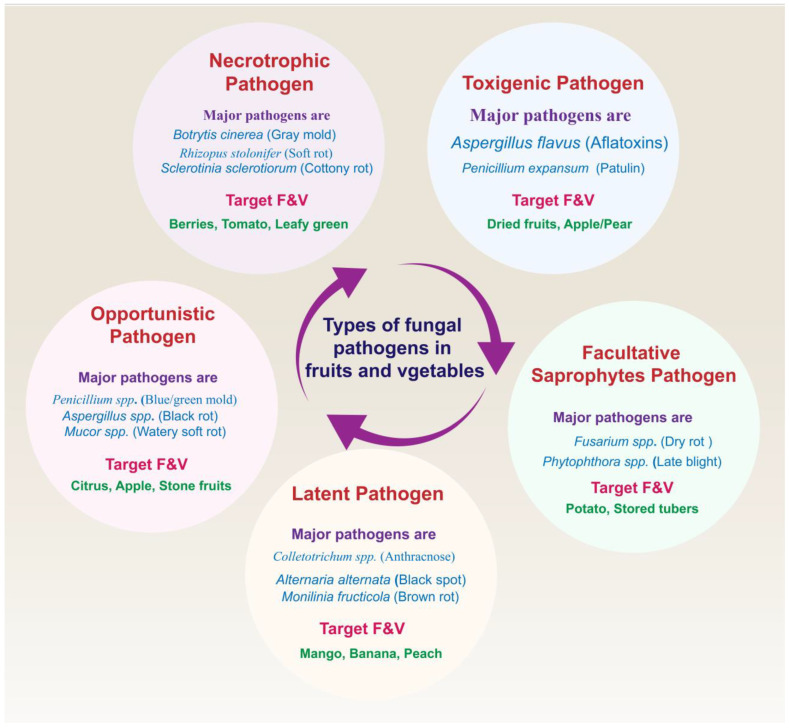
Types of fungal pathogens along with major pathogens and targeted fruits and vegetables.

**Figure 2 foods-14-03075-f002:**
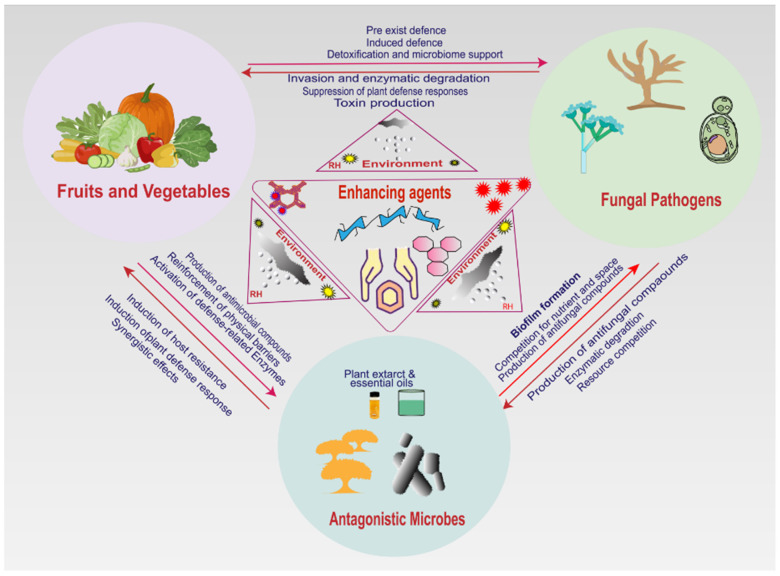
The role of enhancers in interaction with pathogens, antagonistic yeasts, and host cells.

**Figure 3 foods-14-03075-f003:**
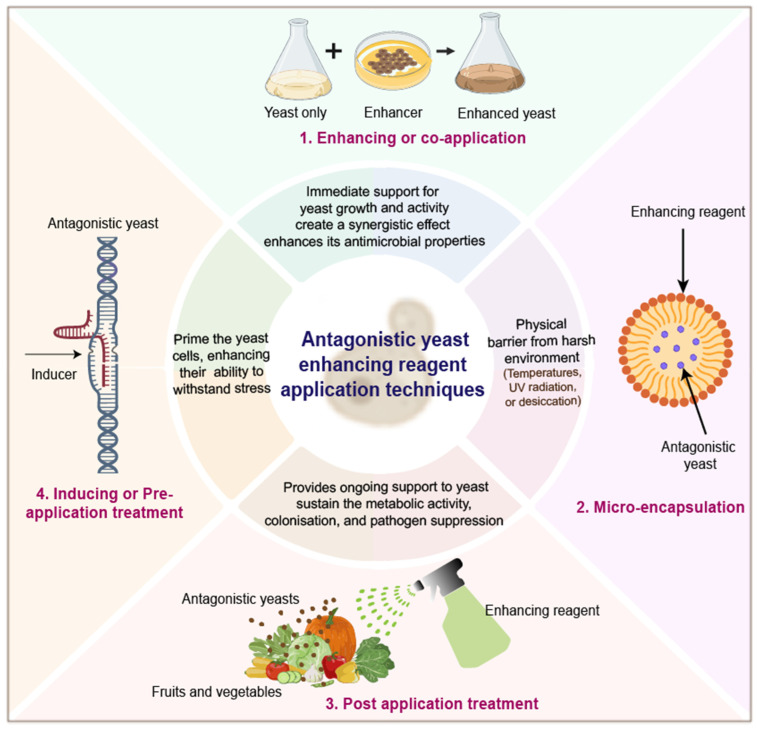
Common action mechanisms of enhancers to improve the performance of antagonistic yeast in controlling fungal pathogens.

**Table 1 foods-14-03075-t001:** Common fungal pathogens in different fruits and vegetables.

No.	Common Fungal Pathogen Species	Fruits and Vegetables Affected	References
1	*Aspergillus niger*, *A. flavus*, *Rhizopus nigra*, *R. oryzae*, *Mucor indicus*, *M. racemosus*, *Candida albicans*, *Penicillium oxalicum*, *P. digitatum*, *Fusarium accuminatum*, *Rhizopus stolonifer*, and *R. nigrican*	Pawpaw	[47]
2	*R. stolonifer*, *A. niger*, *F. accuminatum*, *F. oxysporum*, *F. eqiuseti*, *F. moniliforme*, and *F. solani*	Tomato	
3	*A. niger*, *A. flavus*, *A. fumigatus*, *M. indicus*, *R. nigrican*, *R. nigra*, and *F. accuminatum*	Irish potato	
4	*M. indicus*, *M. amphibiorum*, *M. racemosus*, *A. niger*, *A. flavus*, *A. fumigatus*, *F. accuminatum*, *F. oxysporum*, *R. nigrican*, *R. oligosporus*, and *R. stolonifer*	Carrot	
5	*F. oxysporum*, *F. moniliforme*, *A. niger*, *A. flavus*, *A. fumigatus*, *M. racemosus*, *M. hiemalis*, *C. albicans*, and *P. oxalicum*,	Sweet potato	
6	*F. accuminatum*, *R. stolonifer*, and *A. niger*	Watermelon	
7	*P. expansum*, *M. indicus*, *R. nigrican*, and *F. moniliforme*	Avocado pear	
8	*F. oxysporum*, *F. moniliforme*, *M. indicus*, and *R. nigrican*	Banana and plantain	
9	*F. accuminatum*, *F. moniliforme*, *F. oxysporum*, *A. niger*, *A. flavus*, *A. fumigatus*, *M. indicus*, *M. racemosus*, *M. hiemalis*, *R. nigrican*, and *R. stolonifer*	Palm fruit	
10	*F. oxysporum*, *F. dimerum*, *A. niger*, *M. amphibiorum*, *M. racemosus*, *R. oligosporus*, and *R. stolonifer*	Pepper	
11	*Aspergillus niger*	*Allium sativum* (garlic)	[50]
12	*Aspergillus fumigatus*	*Allium cepa* (onion)	
13	*Alternaria* sp., *Mucor* sp., and *Rhizopus stolonifer*	*Solanum tuberosum* (potato)	
14	*Fusarium* sp.	*Zingiber officinale* (ginger)	
15	*Corynespora* sp.	Solanaceae family, including tomato, chili, brinjal (eggplant), and capsicum (bell pepper)	
16	*Rhizopus* sp.	Tomatoes	
17	*Fusarium* sp.	Chili	
18	*Aspergillus niger*. *Aspergillus flavus,* and *Aspergillus*	Capsicum	
19	*Rhizopus microsporus*	*Phaseolus vulgaris* (common bean)	
20	*Rhizopus* sp.	Carrot	
21	*Fusarium* sp.	Radish, *Luffa acutangular* (ridge gourd), and sapota (sapodilla)	
22	*Fusarium* sp., *Rhizopus* sp., and *Mucor* sp.	Cauliflower	
23	*Rhizopus* sp. and *Mucor* sp.	Papaya	

**Table 2 foods-14-03075-t002:** List of common biocontrol agents used to combat postharvest fruit and vegetable fungal pathogens.

BCA	Species/Subgroups Used	Crop	Pathogen	Refences
Antagonistic Yeast	*Saccharomyces cerevisiae*	Orange	*Penicillium digitatum*	[70]
	*Yarrowia lipolytica*	Apple	*Penicillium expansum*	[71]
	*Hannaella sinensis*	Apple	*Penicillium expansum*	[72]
	*Candida oleophila*	Grape	*Penicillium digitatum*	[73]
	*Candida saitoana*	Apple	*Botrytis cinerea*	[74]
	*Pichia anomala*	Apple	*Botrytis cinerea*	[75]
	*Wickerhamomyces anomalus*	Cherry tomato	*Botrytis cinerea*	[76]
	*Wickerhamomyces anomalus*	Tomato	*Alternaria alternata*	[77]
	*Aureobasidium pullulans S2*	Tomato	*Botrytis cinerea*	[78]
	*Candida sake*	Pear	*Penicillium expansum*, *Botrytis cinerea*	[79]
	*Wickerhamomyces anomalus*	Potato	*A. tenuissima*	[80]
	*Candida utilis*	Tomato	*Alternaria alternata*	[81]
	*Debaryomyce hansenii*	Strawberry	*Rhizopus stolonifer*	[82]
	*Pichia guilliermondii*	Peach	*Penicillium expansum*	[83]
	*Meyerozyma caribbica*	Kiwifruit	*Penicillium expansum*	[84]
	*Meyerozyma guilliermondii*	Kiwifruit	*Botryosphaeria dothidea and Diaporthe actinidiae*	[85]
Antagonistic Bacteria	*Bacillus subtilis*	Apple	*Penicillum expansum*, *Botrytis cinerea*	[86]
	*Pseudomonas fluorescens*	Strawberry	*Botrytis cinerea*	[87]
	*Bacillus amyloliquefaciens*	Peach	*Monilinia fructicola*	[88]
	*Cryptococcus podzolicus*	Pear	*Penicillium expansum*	[89]
	*Lactobacillus plantarum*	Apple	*Penicillium expansum*	[90]
	*Serratia rubidaea* *Mar61-01*	Strawberry	*Botrytis cinerea*	[91]
Essential Oils and Plant Extracts	Different essential oils	In vitro test	*Penicillium italicum*, *Alternaria alternata*	[92]
	Cinnamon oil, ginger oil	Mango	*Colletotrichum gloeosporioides*	[93]
	Cinnamon and clove extract	Tomato	*Alternaria alternata*	[94]
	Neem flower	Peach	*Rhizopus stolonifer and Monilinia fructicola*	[95]
	Neem extract	Tomato	*Diaporthe and Xylaria species*	[96]
	Neem leaf extract	Apple	*Alternaria alternata*, *Aspergillus niger*, *and Penicillium expansum*	[97]
	Neem seed kernel extract	Plum fruit	*Monilinia fructicola*	[98]
	Garlic extract	Mango	*Lasiodiplodia theobromae*	[10]
	Thyme oil	Avocado	*Lasiodiplodia theobromae and* *Colletotrichum gloeosporioides*	[99]

**Table 3 foods-14-03075-t003:** List of common substances used as enhancers, along with their subgroups of BCA, to improve biocontrol effectiveness against pathogens.

Boosting Reagents	Species or Subgroups of BCA Boosted	Pathogens Targeted	Specific Crops	Major Mechanisms of Action	References
Chitosan	*P. anomalus*	*P. expansum*	Grapes	Forms a protective film; induces plant defense; antimicrobial properties	[132]
Chitosan	*R. mucilaginosa*	*B. cinerea*	Strawberries	Increases the abundance of numerous proteins, including ATP synthases and cytochrome C oxidases; reduces levels of ROS; increases abundance and secretion of (CTS2), which could be used to hydrolyze pathogen cell walls	[122]
Phenylethanol	*M. caribbica*	*A. alternata*	Jujube fruit	Chelates metals; enhances BCA survival and biofilm production	[133]
Ascorbic acid	*S. pararoseus Y16*	*P. expansum*	Pears	Antioxidant; reduces oxidative stress; enhances BCA activity	[134]
Ascorbic acid	*P. carribbica*	*P. expansum*	Apples	Antioxidant; reduces oxidative stress; enhances BCA activity	[113]
Trehalose	*S. pararoseus Y16*	*A. carbonarius*	Grapes	Cryoprotection and stabilization; osmotic stress response; antioxidant properties	[135]
Trehalose	*P. carribbica*	*R. stolonifer*, *B. cinerea*	Strawberries	Cryoprotection and stabilization; osmotic stress response; antioxidant properties	[136]
AOS	*D. hansenii*	*P. expansum*	Apples	Activates plant defense; enhances BCA efficacy	[137]
Beta glucans	*S. spartinae W9*	*B. cinerea*	Strawberries	Stimulates plant immune response; enhances BCA colonization	[138]
Salicylic acid	*P. membranaefaciens*	*M. fructicola*	Peaches	Induces systemic acquired resistance (SAR); enhances BCA activity	[139]
L-Methionine	*P. membranaefaciens*	*B. cinerea*	Pears	Nutritional support; enhance antagonistic activity and stress tolerance	[140]
NH_4_Mo	*P. membranaefaciens*	*B. cinerea*	Pears	Direct antifungal activity by interfering with fungal enzyme systems	
Cl_2_Ca	*P. membranaefaciens and V. victoriae*			Enhance stress tolerance in yeast; direct antifungal activity by disrupting fungal cell walls	
L-Serine	*V. victoriae*			Nutritional support for yeast growth; modulate yeast metabolism	
L-Tryptophan	*V. victoriae*			Nutritional support; precursor for antifungal metabolites; may induce plant resistance	
L-Cysteine	*P. membranaefaciens and V. victoriae*	*P. expansum*	Pears	Nutritional support; enhance stress tolerance; modulate sulfur metabolism	
L-Leucine	*P. membranaefaciens and V. victoriae*			Nutritional support for yeast growth and protein synthesis	[140]
NH_4_Mo	*P.membranaefaciens*			Direct antifungal activity by interfering with fungal enzyme systems	
Cl_2_Ca	*P. membranaefaciens and V. victoriae*			Enhance stress tolerance in yeast; direct antifungal activity by disrupting fungal cell walls	
Essential oils (peppermint, melon, and rose oils)	*S. cerevisiae*, *C.tenuis*	*B. cinerea*, *R. stolonifera, and A. alternata*	Tomatoes	Antimicrobial properties; disrupt pathogen cell membranes	[141]
Nutrient additives (amino acids)	*S.cerevisiae*	** *-* **	In vitro	Provide essential nutrients; enhance BCA growth and colonization	[120]
Calcium salts and food-grade antioxidants	*R. glutinis*, *C. laurentii, and A. pullulans*	*P. expansum*	Apples	Stimulation of yeast growth and inhibition of fungal pathogens	[117]

## Data Availability

No data were used in this research.

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
