# Peer review of "Improving Biocontrol Potential of Antagonistic Yeasts Against Fungal Pathogen in Postharvest Fruits and Vegetables Through Application of Organic Enhancing Agents"

_foods, 2025, doi:10.3390/foods14173075_

Round 1

Reviewer 1 Report

Comments and Suggestions for Authors

The authors highlight the role of organic and natural elicitors, their application methods, and action mechanisms in improving BCA overall efficiency. The review summarizes new strategies for optimizing fungal control, reducing food waste, and promoting sustainable agriculture. The topic is interesting, and the ideas are clear. Please see my comments to improve the manuscript.
1. The abbreviation should have its full name when it appears for the first time. What are the full names of GMOs and BCA mentioned in the abstract? Please add it. 
2. The content in lines 38-46 is irrelevant to the main text and should be removed. 
3. Please confirm whether the title in Table 1 is correct. 
4. Please recheck the fourth column of Table 3. It is best to keep the consistent initial capitalization. 
5. There are some problems with the citation formats of references in the manuscript. For example, references (57,58) in Line 238, and [25, 89) in Line 384.
6. Section 3.2 Synthetic fungicides have negative impacts on biodiversity. Please elaborate on what these negative impacts are on biodiversity? 
7. Line 513-514 “[87] in their study reported that co-delivery of the elicitor proves to be an easy and effective method for improvement of agricultural efficiency of the biocontrol yeast”. Please confirm whether this sentence description is correct?
8. Section 6.4 Spray sugar or amino acid supplements on crops pre-treated with yeast. How they play a role, please add. 
9. It is suggested that the outlook for the future be supplemented in the conclusion section.

Author Response

Dear editor,

We greatly appreciate the time and effort you and the reviewers devoted to providing valuable comments on our manuscript. The feedback and suggestions helped us to improve the quality and content of our manuscript. The major changes made in the revised manuscript are marked in red color. Here we respond to the reviewer’s comment point by point.

Referee: 1
Comments and Suggestions for Authors

The authors highlight the role of organic and natural elicitors, their application methods, and action mechanisms in improving BCA overall efficiency. The review summarizes new strategies for optimizing fungal control, reducing food waste, and promoting sustainable agriculture. The topic is interesting, and the ideas are clear. Please see my comments to improve the manuscript.

Dear reviewer, Thank you for taking your precious time to review our work, and your feedback helps us to improve the quality and content of our manuscript.

1.  The abbreviation should have its full name when it appears for the first time. What are the full names of GMOs and BCA mentioned in the abstract? Please add it. 

Dear reviewer, Thank you for the suggestion. We defined the mentioned abbreviations and other abbreviations in the manuscript for the first time they were used.

  1. The content in lines 38-46 is irrelevant to the main text and should be removed.

Dear reviewer, Thank you for the suggestion. You are right; the content is irrelevant, and it was included unintentionally. We revised the content according to your suggestion.

  1. Please confirm whether the title in Table 1 is correct.

Dear reviewer, Thank you for the suggestion. The title has been revised now.

  1. Please recheck the fourth column of Table 3. It is best to keep the consistent initial capitalization. 

Dear reviewer, Thank you for the suggestion. We revised the names in initial capitalization as suggested in our revised manuscript

  1. There are some problems with the citation formats of references in the manuscript. For example, references (57,58) in Line 238, and [25, 89) in Line 384.

Dear reviewer, Thank you for the suggestion. Both the citation formats are now revised.

  1. Section 3.2 Synthetic fungicides have negative impacts on biodiversity. Please elaborate on what these negative impacts on biodiversity are.

Dear reviewer, Thank you for the excellent suggestion. We included the elaboration for stated negative impacts of synthetic fungicides on biodiversity.

  1. Line 513-514 “[87] in their study reported that co-delivery of the elicitor proves to be an easy and effective method for improvement of agricultural efficiency of the biocontrol yeast.” Please confirm whether this sentence description is correct.

Dear reviewer, Thank you for pointing out an important question. According to this sentence, the researchers found a way to make a beneficial yeast (a natural, living pesticide) work even better. They did this by packaging the yeast together with a special "booster" chemical (the elicitor) and applying them to the plants at the same time and concluded that this method is both easy to do and highly effective at improving crop health and yield. Further description was added to avoid ambiguity. Thank you!

  1. Section 6.4 Spray sugar or amino acid supplements on crops pre-treated with yeast. How they play a role, please add. 

Dear reviewer, Thank you for the suggestion. The description how this pretreatment play role in improving biocontrol efficacy of yeast added in our revised manuscript

  1. It is suggested that the outlook for the future be supplemented in the conclusion section.

Dear reviewer, Thank you for the excellent suggestion. We included the future outlook studies as per your suggestion.

Reviewer 2 Report

Comments and Suggestions for Authors

The topic of this work (Postharvest biocontrol of fruits and vegetables fungal diseases with antagonistic yeasts) has been widely researched, so I consider that the present study is not so original. However, this work is interesting because it addresses and documents with reliable, relevant and up-to-date references  topics of great interest, such as: Causes of postharvest loss in fruit and vegetables; Common postharvest diseases of fruit & vegetables and their fungal disease management (physical, chemical and biological control methods); Factors affecting the efficacy of antagonistic microbes; Common reagents used as enhancer of antagonistic yeast and their characteristic; Application methods of enhancer on antagonistic yeast performance; Mechanism involved in enhancement. Also shows that inducing, enhancing, co-application, encapsulation, and post-application treatment are common enhancement techniques while environmental, host and pathogen characteristics, antagonistic microbial traits, and chemical inputs are the major gearing factors for the best application methods.

Besides, the study highlights the role of organic and 30 natural elicitors, their application methods, and action mechanisms in improving BCA overall efficiency. This review concisely compiles recent strategies alerting for further research to revolutionize fungal pathogen management, reduce food waste, and promote responsible farming practices. Also focuses on three key areas; i) To trigger the importance and necessity of improving antagonistic yeast,: identifying and compiling reagents utilized in several study to improve/boost the performance; ii) highlights the possible application methods: exploring effective delivery techniques of enhancers along with antagonistic yeasts to mitigate target fungal pathogens and; iii) compiling different mechanisms of actions: exposing the underlying mechanisms behind the various improvements of antagonistic yeasts against the development of pathogenic fungi.

This review concludes by emphasizing that enhancing the efficacy of antagonistic microbes has emerged as a viable option for overcoming limitations such as hindered by reduced effectiveness, product specificity, variability at different environmental conditions, short shelf life, limited mass production, less accessibility for market demand, and low awareness on its action mechanisms. It also concludes that an array of organic substances and natural constituents, such as nutrient supplements, antioxidants, and growth regulators, have proven successful in improving the efficacy of biocontrol techniques.

The manuscript is clearly written, properly supported with relevant and up-to-date references and duly discussed and concluded. 

However, below I indicate some observations that must be satisfactorily addressed by the authors.

On lines 1032-1034 in the References section, reference 159 is apparently not cited in the manuscript. Finally, I also propose to the authors that the title of their work be more precise by changing the last part, so that it could be any of the following alternatives:

  1. Improving Biocontrol Potential of Antagonistic Yeasts Against Fungal Pathogen in Postharvest Fruits and Vegetables Through Application of Organic Inputs, or
  2. Improving Biocontrol Potential of Antagonistic Yeasts Against Fungal Pathogen in Postharvest Fruits and Vegetables Through Application of Organic Compounds, or
  3. Improving Biocontrol Potential of Antagonistic Yeasts Against Fungal Pathogen in Postharvest Fruits and Vegetables Through Application of Organic Roosting Reagents, or
  4. Improving Biocontrol Potential of Antagonistic Yeasts Against Fungal Pathogen in Postharvest Fruits and Vegetables Through Application of Organic Enhancing Agents

Author Response

Dear editor,

We greatly appreciate the time and effort you and the reviewers devoted to providing valuable comments on our manuscript. The feedback and suggestions helped us to improve the quality and content of our manuscript. The major changes made in the revised manuscript are marked in red color. Here we respond to the reviewer’s comment point by point

Referee: 2
Comments and Suggestions for Authors

The topic of this work (Postharvest biocontrol of fruits and vegetables fungal diseases with antagonistic yeasts) has been widely researched, so I consider that the present study is not so original. However, this work is interesting because it addresses and documents with reliable, relevant and up-to-date references  topics of great interest, such as: Causes of postharvest loss in fruit and vegetables; Common postharvest diseases of fruit & vegetables and their fungal disease management (physical, chemical and biological control methods); Factors affecting the efficacy of antagonistic microbes; Common reagents used as enhancer of antagonistic yeast and their characteristic; Application methods of enhancer on antagonistic yeast performance; Mechanism involved in enhancement. Also shows that inducing, enhancing, co-application, encapsulation, and post-application treatment are common enhancement techniques while environmental, host and pathogen characteristics, antagonistic microbial traits, and chemical inputs are the major gearing factors for the best application methods.

Besides, the study highlights the role of organic and 30 natural elicitors, their application methods, and action mechanisms in improving BCA overall efficiency. This review concisely compiles recent strategies alerting for further research to revolutionize fungal pathogen management, reduce food waste, and promote responsible farming practices. Also focuses on three key areas; i) To trigger the importance and necessity of improving antagonistic yeast,: identifying and compiling reagents utilized in several study to improve/boost the performance; ii) highlights the possible application methods: exploring effective delivery techniques of enhancers along with antagonistic yeasts to mitigate target fungal pathogens and; iii) compiling different mechanisms of actions: exposing the underlying mechanisms behind the various improvements of antagonistic yeasts against the development of pathogenic fungi.

This review concludes by emphasizing that enhancing the efficacy of antagonistic microbes has emerged as a viable option for overcoming limitations such as hindered by reduced effectiveness, product specificity, variability at different environmental conditions, short shelf life, limited mass production, less accessibility for market demand, and low awareness on its action mechanisms. It also concludes that an array of organic substances and natural constituents, such as nutrient supplements, antioxidants, and growth regulators, have proven successful in improving the efficacy of biocontrol techniques.

The manuscript is clearly written, properly supported with relevant and up-to-date references and duly discussed and concluded. 

However, below I indicate some observations that must be satisfactorily addressed by the authors.

On lines 1032-1034 in the References section, reference 159 is apparently not cited in the manuscript. Finally, I also propose to the authors that the title of their work be more precise by changing the last part, so that it could be any of the following alternatives:

  1. Improving Biocontrol Potential of Antagonistic Yeasts Against Fungal Pathogen in Postharvest Fruits and Vegetables Through Application of Organic Inputs, or
  2. Improving Biocontrol Potential of Antagonistic Yeasts Against Fungal Pathogen in Postharvest Fruits and Vegetables Through Application of Organic Compounds, or
  3. Improving Biocontrol Potential of Antagonistic Yeasts Against Fungal Pathogen in Postharvest Fruits and Vegetables Through Application of Organic Boosting Reagents, or
  4. Improving Biocontrol Potential of Antagonistic Yeasts Against Fungal Pathogen in Postharvest Fruits and Vegetables Through Application of Organic Enhancing Agents

Dear reviewer thank you very much for your excellent comments, suggestions, and concern. We appreciate your deep-down observation of our study. Thank you for your excellent observation of reference 159. It was a typing error and is now corrected and apparently cited in the table. We really value your thoughts and agree with all the comments and suggestions. Your 4th suggestion for our manuscript title is really attractive and meaningful, and we changed the topic to this one.

Reviewer 3 Report

Comments and Suggestions for Authors

The topic of biocontrol in the postharvest of fruits and vegetables is innovative and aligns with the current trend of studying alternatives to conventional fungicides. It is also relevant given the limited number of fungicidal products approved for postharvest application on fruits and vegetables.

Lines 38–46: Remove the instructions that remain from the journal template.
Lines 52–53: The statement “Significant research and innovation efforts have focused on reducing F&V postharvest losses” could be expanded by indicating which actions have been implemented to address these losses.

As this is a review, the authors should include a paragraph describing the methodology used for the literature search. This should specify the databases consulted, whether any artificial intelligence tools were employed, the keywords used, and whether the search was restricted to a given time period (e.g., the last 10 years).

Lines 187–195: Scientific names are incorrectly formatted. Please ensure consistency throughout the manuscript, with the genus capitalized and the species in lowercase.

Table 1: The table presents studies with different matrices but cites only two sources, one of which appears to be a review. Ideally, the original sources of the information should be referenced.

Table 2: It is recommended to add a column for each study indicating the treatment effects (positive or negative), the extent of reduction in phytopathogenic fungal growth, or overall efficacy.

In section “4.4. Chemical inputs”, beyond pesticides, the authors could also consider whether postharvest treatments such as hypochlorite, peracetic acid, or neutral soaps may interfere with the action of antagonistic microbes, since these products may leave residues on the plant matrix.

Table 3: Ensure consistency in text formatting. Currently, there are words in bold and variations in capitalization.

Lines 460–461: In the sentence “According to study conducted by [93] precultured or induced yeasts increases the induc…,” please include the author’s name (year) before the reference number [93].

Figure 2: Correct the spelling errors in the image text (e.g., “extarct,” “campaounds”) and check for others.

Line 592: Replace “yeas” with the correct form.

Author Response

Dear editor,

We greatly appreciate the time and effort you and the reviewers devoted to providing valuable comments on our manuscript. The feedback and suggestions helped us to improve the quality and content of our manuscript. The major changes made in the revised manuscript are marked in red color. Here we respond to the reviewer’s comment point by point

Referee:3
Comments and Suggestions for Authors

The topic of biocontrol in the postharvest of fruits and vegetables is innovative and aligns with the current trend of studying alternatives to conventional fungicides. It is also relevant given the limited number of fungicidal products approved for postharvest application on fruits and vegetables.

Dear reviewer thank you for excellent observation. We value your comments

Lines 38–46: Remove the instructions that remain from the journal template.

Dear reviewer, thank you for your comments; the instruction was removed from the revised manuscript

Lines 52–53: The statement “Significant research and innovation efforts have focused on reducing F&V postharvest losses” could be expanded by indicating which actions have been implemented to address these losses.

Dear reviewer, thank you for your suggestion. The statement was expanded and explained in our revised manuscript

As this is a review, the authors should include a paragraph describing the methodology used for the literature search. This should specify the databases consulted, whether any artificial intelligence tools were employed, the keywords used, and whether the search was restricted to a given time period (e.g., the last 10 years).

Dear reviewer, Thank you for your excellent suggestion. The suggested methodology includes how the literature in this review manuscript was accessed, including the key terminologies used to search.

Lines 187–195: Scientific names are incorrectly formatted. Please ensure consistency throughout the manuscript, with the genus capitalized and the species in lowercase.

Dear reviewer, Thank you for the excellent suggestion. The names are corrected in the specified line and entire revised manuscript

Table 1: The table presents studies with different matrices but cites only two sources, one of which appears to be a review. Ideally, the original sources of the information should be referenced.

Dear reviewer, Thank you for the suggestion. the original reference cited as per your comments

Table 2: It is recommended to add a column for each study indicating the treatment effects (positive or negative), the extent of reduction in phytopathogenic fungal growth, or overall efficacy.

Dear reviewer, thank you for the excellent suggestion. The entire table of content is about the effects of common biocontrol agents, which have a negative impact on pathogens while they havepositive effects on fruits and vegetables or host cells. The extent of reduction in fungal pathogens will be very detailed and wordy, while the major ones are described in the explanation parts of the manuscript.

In section “4.4. Chemical inputs,” beyond pesticides, the authors could also consider whether postharvest treatments such as hypochlorite, peracetic acid, or neutral soaps may interfere with the action of antagonistic microbes, since these products may leave residues on the plant matrix.

Dear reviewer, thank you for your nice suggestion. The description is included in the revised manuscript.

Table 3: Ensure consistency in text formatting. Currently, there are words in bold and variations in capitalization.

Dear reviewer, thank you for your suggestions. The stated words are corrected in accordance with others.

Lines 460–461: In the sentence “According to a study conducted by [93] precultured or induced yeasts increases the induc…,” please include the author’s name (year) before the reference number [93].

Dear reviewer, thank you for your suggestion. it was corrected as per your suggestion

Figure 2: Correct the spelling errors in the image text (e.g., “extarct,” “campaounds”) and check for others.

Line 592: Replace “yeas” with the correct form.

Dear reviewer, thank you for excellent observation. the spelling error corrected in revised manuscript

Reviewer 4 Report

Comments and Suggestions for Authors

16 August, 2025

Manuscript ID: foods-3784327

Title of the manuscript: Improving Biocontrol Potential of Antagonistic Yeasts Against Fungal Pathogen in Postharvest Fruits and Vegetables Through Application of Organic Constituents

The study provides readers with comprehensive information on whether biocontrol can be used as a method to prevent postharvest losses in fruits and vegetables. The authors have reviewed and presented the latest research. I recommend expanding some sections of the text. Also, the author has made numerous self-citations. The number of these citations should be reduced. I have detailed my comments in the attached file. I believe the study will contribute to food science. The study need minor revisions.

Note: My minor suggestions were shown on the annotated PDF file.

With my best regards

Author Response

Dear editor,

We greatly appreciate the time and effort you and the reviewers devoted to providing valuable comments on our manuscript. The feedback and suggestions helped us to improve the quality and content of our manuscript. The major changes made in the revised manuscript are marked in red color. Here we respond to the reviewer’s comment point by point

Referee: 4
Comments and Suggestions for Authors

The study provides readers with comprehensive information on whether biocontrol can be used as a method to prevent postharvest losses in fruits and vegetables. The authors have reviewed and presented the latest research. I recommend expanding some sections of the text. Also, the author has made numerous self-citations. The number of these citations should be reduced. I have detailed my comments in the attached file. I believe the study will contribute to food science. The study need minor revisions.

Dear reviewer, thank you for you devoted time and effort to review our manuscript. We have carefully reviewed the references you highlighted. While many are basic to our manuscript's scholarly content and remain essential, we have revised the list by incorporating more recent studies and have diversified the sources to include work from different laboratories to mitigate potential bias.

Causes of postharvest loss in fruit and vegetables: this section need expand

We appreciate the reviewer's feedback. While we agree the section should cover all important points, we have condensed the discussion to comply with word count constraints. The current content captures the essential information effectively.

A table should be provided showing which synthetic fungicides are used and which fruit are most commonly used on. A few research findings should be presented.

Dear reviewer thank you for the suggestion. While we appreciate the value of additional tables, the primary focus of this study is to improve the efficacy of antagonistic yeasts as biocontrol agents. Chemical methods are discussed primarily to provide context and a smooth narrative flow for this main objective.

  1. Factors deterring the efficacy of antagonistic microbes

Dear reviewer, thank you for your suggestion. the word "affects" is replaced by "determining" according to your suggestion

Please no bold

Dear reviewer, Thank you for your suggestion. the bold words are corrected consistently at the specified comments and in the entire revised manuscript
